# The Role of In-Group and Out-Group Facial Feedback in Implicit Rule Learning

**DOI:** 10.3390/bs13120963

**Published:** 2023-11-23

**Authors:** Meijun Ou, Wenjie Peng, Wenyang Zhang, Muxin Ouyang, Yiling Liu, Keming Lu, Xiangyan Zeng, Jie Yuan

**Affiliations:** 1School of Psychology, South China Normal University, Guangzhou 510631, China; 2020023444@m.scnu.edu.cn (M.O.); 2023023873@m.scnu.edu.cn (W.P.); 20182921043@m.scnu.edu.cn (W.Z.); 2021023704@m.scnu.edu.cn (Y.L.); 2021023798@m.scnu.edu.cn (K.L.); zengxy@scnu.edu.cn (X.Z.); 2Psychology Department, Skidmore College, Saratoga Springs, NY 12866, USA; muxinouyang@skidmore.edu

**Keywords:** implicit learning, social feedback, facial expression, in-group, out-group

## Abstract

Implicit learning refers to the fact that people acquire new knowledge (structures or rules) without conscious awareness. Previous studies have shown that implicit learning is affected by feedback. However, few studies have investigated the role of social feedback in implicit learning concretely. Here, we conducted two experiments to explore how in-group and out-group facial feedback impact different difficulty levels of implicit rule learning. In Experiment 1, the Chinese participants in each group could only see one type of facial feedback, i.e., either in-group (East Asian) or out-group (Western) faces, and learned the implicit rule through happy and sad facial expressions. The only difference between Experiment 2 and Experiment 1 was that the participants saw both the in-group and out-group faces before group assignment to strengthen the contrast between the two group identities. The results showed that only in Experiment 2 but not Experiment 1 was there a significant interaction effect in the accuracy of tasks between the difficulty levels and groups. For the lowest difficulty level, the learning accuracy of the in-group facial feedback group was significantly higher than that of the out-group facial feedback group, whereas this did not happen at the two highest levels of difficulty. In conclusion, when the contrast of group identities was highlighted, out-group feedback reduced the accuracy of the least difficult task; on the contrary, there was no accuracy difference between out-group and in-group feedback conditions. These findings have extensively important implications for our understanding of implicit learning and improving teaching achievement in the context of educational internationalization.

## 1. Introduction

### 1.1. Implicit Learning

Implicit learning refers to individuals acquiring complex knowledge (structures or rules) from stimulus but not being aware of or able to articulate the knowledge learned [1,2]. Compared to explicit learning, implicit learning is an automatic learning process without intention or conscious hypothesis testing [3,4,5,6]. For example, when individuals were asked to memorize a set of letter strings generated by an artificial grammar, they became increasingly sensitive to the grammatical structure of the stimuli though they were unable to verbalize the rules of the grammar [2].

Implicit rule learning, such as learning sequential and non-sequential structures, is an essential part of implicit learning. Many studies have demonstrated that individuals can learn rules implicitly in various domains using different implicit rule-learning tasks. People have been found to be able to unconsciously learn visual sequences [7], semantic structures [8], and non-sequential functions in a video game [9] and discriminate a rule consistent from rule-violating card triads [10]. However, most studies have focused on the overall task performance but fewer have concentrated on how different task difficulties impact individuals’ performance.

### 1.2. Effect of Social Feedback on Implicit Rule Learning

Feedback is one of the factors that strongly influences abstract learning [11,12,13,14,15,16]. Nevertheless, little work has focused on the effect of social feedback information in implicit rule-learning environments. Only one study began to compare the effect of social and non-social feedback on implicit rule learning [10]. The participants in the study were asked to discriminate between the card triads based on whether the card triads complied with a rule. The rule was given only by social or non-social feedback. Social feedback was a happy, neutral, or sad facial expression, and non-social feedback was a traffic light icon. These authors found that social feedback interfered with the participants’ learning. However, it is not enough to study the impact of social feedback by focusing solely on comparing the effects of social and non-social feedback. Because different types of social feedback may yield different results in implicit rule learning, this study was expected to deepen the understanding of the specific role of social feedback.

### 1.3. In-Group and Out-Group Facial Feedback

Facial expression is one of the common types of social feedback that can signal a correct or incorrect answer with a smile or frown [10,17,18,19,20]. According to the group members’ information, facial expression feedback can be divided into two types: in-group facial feedback and out-group facial feedback. In-group facial feedback refers to the facial expression of individuals within the same ethnic group as the observer. Out-group facial feedback refers to the facial expression of individuals from different ethnic groups than the observer. For example, for Chinese people, the feedback from East Asian faces could be in-group facial feedback, and western individuals’ facial expressions could be out-group facial feedback. 

In recent decades, the coexistence of in-group and out-group facial feedback has become more and more common and important in learning areas under the trend of educational internationalization in China, as well as worldwide. With the growing number of foreign teachers, and accessibility of educational resources around the world as well as Internet platforms, more and more people, especially students, can receive facial feedback from both in-group and out-group teachers when they implicitly acquire knowledge. Given that the group membership information influences people’s perceptual awareness [21], perceptual preferences [22], memory [23], and affective judgments [24] of others’ faces, the difference between in-group and out-group facial feedback may be crucial when people learn with facial feedback. However, little is known about how the in-group and out-group facial feedback affect individual learning, especially implicit rule learning. 

In addition, a previous study suggested that cognitive busyness may lead participants to categorize others only based on the dimensions directly relevant to their current goals of cognitive tasks rather than on their stereotype [25]. According to that, people would be likely to pay more attention to the feedback information of the facial expression than the characteristic of its group membership in an implicit learning task if they were not reminded that the faces represented different group members. Thus, we expected to know whether comparing in-group and out-group feedback directly influenced the performance of implicit learning, so that we could learn more about the relationship between social feedback and implicit learning. 

### 1.4. The Current Study

Beston et al. compared the influence of social versus non-social feedback on implicit rule learning [10]. Based on this, the current study further examined how in-group and out-group facial feedback influenced implicit rule learning. It is worth noting that we also distinguished between the different difficulty levels of implicit rule learning to explore how participants perform under different cognitive demands. Specifically, we presented two experiments using an implicit–intentional learning task adapted from Beston et al. [10] and conducted 2 × 4 mixed-design ANOVAs with East Asian and Western facial expression feedback conditions as between-subject factors and difficulty levels as within-subject factors. Experiment 1 aimed to assess how East Asian and Western facial feedback would shape implicit rule-learning task performance of different difficulty levels without the reminder of the group identity of the feedback. The only difference between Experiment 2 and Experiment 1 was that the participants saw both the in-group and out-group faces before group assignment to strengthen the contrast between the two group identities. We predicted that facial feedback would affect or partially affect the participants’ learning performance in both experiments. The current study contributes to a better understanding of the influencing factors and mechanisms of implicit rule learning. Moreover, it has implications for maximizing the effect of teaching and learning in an international educational environment. 

## 2. Methods

### 2.1. Participants

Experiment 1 recruited 64 Chinese undergraduate students from South China Normal University. The participants were recruited via online advertisements with a link to a sign-up survey. Online advertisements were distributed on social media platforms (e.g., WeChat groups). Then, the researchers contacted the eligible participants according to their contact information and available time schedules reported in the sign-up survey. Due to technical issues, we eliminated ten incomplete individual datasets. Thus, the final sample for statistical analysis consisted of 54 participants in the learning phase (27 in-group, 23 females; 27 out-group, 21 females) and 54 participants in the test phase (26 in-group, 23 females; 28 out-group, 22 females). It must be noted that one male participant in the in-group facial feedback condition and one female participant in the out-group facial feedback condition completed both the learning phase and the test phase, but the datasets of the testing phase of the male participant and the learning phase of the female participant were missing due to procedural errors. As with the method of Beston et al. [10], we included the existing data of these two participants in the final statistical analysis. The mean age was 20.50 years (*SD* = 1.92).

A total of 62 students from South China Normal University were recruited in Experiment 2 following the same routine as in Experiment 1. After excluding incomplete individual datasets caused by technical issues, the final sample for statistical analysis in the learning phase consisted of 51 individuals (25 in-group, 19 females; 26 out-group, 21 females) and 50 individuals (24 in-group, 18 females; 26 out-group, 21 females) in the test phase. One female participant in the in-group facial feedback condition completed both phases, but her dataset in the testing phase was missing due to procedurals errors. We also included the existing data of this participant in the final statistical analysis. The mean age was 19.85 (*SD* = 2.20) years old. 

All Participants signed up voluntarily for the experiment and were compensated for their participation. The South China Normal University’s ethics committee authorized all research procedures, including Experiments 1 and 2, and the participants provided written informed consent prior to their participation. The participants were randomly assigned to either the in-group or the out-group in the facial feedback condition. To prevent confounding variables, all participants had no professional training in psychology and had never participated in similar experiments. The participants in the study were in good mental health (no psychiatric or neurological disorders), right-handed, colorblindness-free, and had normal or corrected-to-normal vision. 

### 2.2. Apparatus and Stimuli

The experiment was designed and run using E-Prime 2.0. The visual stimulus was displayed in the middle of a 24-inch CRT monitor (1920 × 1080) with a refresh rate of 100 Hz. The viewing distance was about 70 cm.

We adapted the implicit–intentional learning task from Beston et al. [10] as the stimulus because this task both assessed the overall performance of implicit rule learning and allowed us to explore the role of social feedback at different difficulty levels. The stimulus for the implicit–intentional learning task included different combinations of card triads (i.e., three cards in each trial). The figures in each card contained four dimensions: shape (S: circle, square, and triangle), color (C: red, green, and blue), number (N: one, two, and three), and filling (F: empty, hashed, and full). Based on the combination of the four dimensions, a total of eighty-one cards were created which then composed the different card triads. If the elements of each dimension (S, C, N, and F) of all the cards were the same or different in a card triad, the triad of cards was considered to be a legal card triad complying with the implicit rule. In other words, a card triad would be illegal if any dimension of the three cards was not completely identical or different. Moreover, considering that different difficulties caused by the complex change of card triads could affect the process of implicit learning, the card triads were divided into four difficulty levels. The creation of the card triads and the randomization of trials in each difficulty level were implemented entirely by programming with E-Prime 2.0. Examples of legal and illegal card triads with different difficulty levels are presented in Figure 1. 

Images of facial expressions served as the stimulus for facial feedback. The photographs were chosen from the NimStim Set of Facial Expressions [26] (http://www.macbrain.org/resources.htm, accessed on 8 June 2022). The NimStim Set is a multiracial face set containing a variety of expressions, providing easy access to images of both East Asian and Western faces. Thus, this set was an appropriate choice to obtain in-group and out-group facial feedback. However, most of the East Asian facial images in the stimulus set were of East Asian females. There was only one image in the East Asian male stimulus set. Furthermore, more than half of elementary and secondary school teachers are female in China [27]. Thus, we chose three groups of facial images of East Asian and Western females. Each female’s facial image displayed both positive (happy) and negative (sad) facial expressions. We chose the happy expressions with open mouths and sad expressions with closed mouths. Twelve images were used as facial feedback stimuli in total. They were presented with either positive or negative facial expressions.

In the practice phase, the card trials and facial expressions were different from those in the experimental phase. In other words, if a card triad or a facial image was used in the practice phase, it would not appear in the experimental phase. Thus, there were only two female facial images in the experiment phase.

### 2.3. Procedure

After the completion of the informed consent form, there were eight practice trials to familiarize the participants with the experimental procedure. The participants then completed the formal experiments, including the learning and test phase, in the lab environment.

#### 2.3.1. Procedure of Experiment 1

Learning phase. During the learning phase, the participants initially conducted an implicit intentional learning task designed by Beston et al. [10]. Before starting Experiment 1, the participants read an instruction with guidance on how to complete the task. Specifically, the participants were instructed to press the designated keys to indicate whether the card triads complied with a rule, i.e., whether they were either legal or illegal, and adjust their judgment on the basis of the facial feedback they saw. In the instructions of each condition, the participants were presented with only one kind of facial feedback, either in-group or out-group facial feedback. In the experiment, cross-fixations were used to direct the participant’s attention to the screen’s center. Both legal and illegal card triads had the same probability and presentation difficulty (1:1). After a red fixation point ‘+’ lasting 200 ms, a random card triad was presented in each trial. The participants identified the legality of the card triad. The feedback was then presented for 750 ms. As all the participants were Chinese, those in the in-group facial feedback group received feedback from images of East Asian faces, and those in the out-group facial feedback group were shown images of Western faces as feedback. In both groups, the face showed a happy expression when a response was correct and a sad expression when it was incorrect (see Figure 2). Using a stair-like procedure, the participants learned four progressively challenging learning levels. In other words, before achieving five correct responses, any error would reset the number of correct trials to zero. After the participants made five correct cumulative judgments for a difficulty level in each legal and illegal condition, they moved on to the next difficulty level to wrap up their learning. After 200 trials at a given difficulty level, if the cumulative number of correct judgments did not reach five, the participants automatically advanced to the next difficulty level. Moreover, the response side was counterbalanced across the participants.

Test phase. The participants were asked to complete the same task in the test phase as in the learning phase. Legal and illegal card triads were presented with equal probability and difficulty (1:1). However, the participants were instructed to make a judgment within 1500 ms, and there was no facial feedback in the test phase. Three blocks of 200 trials were accomplished by each participant and there was a one-minute break between each block. The card triads of the test phase had never previously appeared in the learning phase. The presentation of the card triads was arbitrary and did not take the difficulty levels into account. The experimental procedure for the test phase is depicted in Figure 3. 

#### 2.3.2. Procedure of Experiment 2

The instructions and procedures of Experiment 2 were identical to those in Experiment 1 except for one critical difference. That is, the participants in the in-group facial feedback condition and out-group facial feedback condition were shown images of both East Asian and Western faces prior to the start of group assignment to perceive the contrast between in-group and out-group faces. 

### 2.4. Data Analysis

Both Experiment 1 and Experiment 2 used 2 × 4 mixed-design ANOVAs with facial feedback conditions (in-group and out-group) as between-subject factors and difficulty levels (level 1, level 2, level 3, and level 4) as within-subject factors. The dependent measures were the participants’ task performance, including the reaction times (RTs) of correct responses and accuracy (ACC). To avoid biases in the study, we excluded the responses with reaction times less than 200 ms or above 2.5 standard deviations of the average. Bonferroni correction was applied to all post hoc comparisons.

## 3. Results

### 3.1. Experiment 1 Results

#### 3.1.1. Learning Phase

Accuracy. There was no main effect of facial feedback conditions in terms of accuracy, *F* (1, 52) = 1.392, *p =* 0.243, nor a significant interaction effect between the facial feedback conditions and difficulty levels, *F* (3, 156) = 0.351, *p* = 0.789. However, we found a significant main effect of difficulty level, *F* (3, 156) = 19.998, *p* < 0.001, *η^2^_p_* = 0.278. Bonferroni post hoc comparisons indicated that the accuracy of level 1 (*M*_level 1_ = 0.863, *SD =* 0.124) was significantly higher than that of level 2 (*M*_level 2_ = 0.805, *SD =* 0.101, *t* (53) = 2.999, *p* = 0.026), level 3 (*M*_level 3_ = 0.719, *SD =* 0.151, *t* (53) = 6.155, *p* < 0.001), and level 4 (*M*_level 4_ = 0.726, *SD =* 0.127, *t* (53) = 5.540, *p* < 0.001). The accuracy of level 2 was also significantly higher than that of level 3 (*t* (53) = 4.611, *p* < 0.001) and level 4 (*t* (53) = 3.818, *p =* 0.002), but there was no significant difference between the accuracy of level 3 and level 4 (*t* (53) = −0.294, *p* = 1.000). Overall, from level 1 to level 4, the accuracy gradually decreased. 

Reaction times. The results showed that there was also no significant group difference in the facial feedback conditions for the reaction times, *F* (1, 52) = 1.735, *p* = 0.194, nor a significant interaction effect, *F* (3, 156) = 0.978, *p* = 0.405. Nevertheless, there was a significant main effect of difficulty level, *F* (3, 156) = 4.462, *p =* 0.005, *η^2^_p_* = 0.079. Bonferroni post hoc comparisons showed that the reaction times of level 2 (*M*_level 2_ = 1228, *SD* = 452) were significantly slower than those of level 3 (*M*_level 3_ = 1404, *SD* = 701, *t* (53) = −3.066, *p* = 0.020) and slightly slower than those of level 4 (*M*_level 4_ = 1390, *SD* = 548, *t* (53) = −2.744, *p* = 0.053). However, there was no significant difference in reaction times between level 1 (*M*_level 1_ = 1227, *SD* = 462) and level 2 (*t* (53) = −0.013, *p* = 1.000), nor between level 3 and level 4 (*t* (53) = 0.185, *p* = 1.000). 

#### 3.1.2. Test Phase

Accuracy. Analysis in the test phase showed that there was no significant main effect of the facial feedback conditions in terms of accuracy, *F* (1, 52) = 0.297, *p =* 0.588 (see Figure 4), nor an interaction, *F* (3, 156) = 1.226, *p* = 0.302, whereas a significant main effect of difficulty level was found, *F* (3, 156) = 13.795, *p* < 0.001, *η^2^_p_* = 0.210. Bonferroni post hoc comparisons showed that the accuracy of level 1 (*M*_level 1_ = 0.742, *SD =* 0.147) was significantly higher than the accuracy of the other three levels (*M*_level 2_ = 0.696, *SD =* 0.121, *t* (53) = 4.106, *p* = 0.001; *M*_level 3_ = 0.673, *SD =* 0.098, *t* (53) = 3.486, *p* = 0.006; *M*_level 4_ = 0.607, *SD =* 0.156, *t* (53) = 4.084, *p* = 0.001). The accuracy of level 3 was also significantly higher than that of level 4 (*t* (53) = 3.735, *p* = 0.002). However, there was no significant difference between the accuracy of level 2 and level 3 (*t* (53) = 1.813, *p* = 0.401). Similar to the learning phase, accuracy gradually decreased from level 1 to level 4.

Reaction times. There was also no main effect of the facial feedback conditions for the reaction times, *F* (1, 52) = 0.189, *p* = 0.184, nor an interaction, *F* (3, 156) = 0.379, *p* = 0.768. But there was a significant main effect of difficulty level, *F* (3, 156) = 21.649, *p* < 0.001, *η^2^_p_* = 0.294. Bonferroni post hoc comparisons indicated that the reaction times of level 1 (*M*_level 1_ = 848, *SD =* 105) were significantly faster than the reaction times of the other three levels (*M*_level 2_ = 875, *SD* = 104, *t* (53) = −5.332, *p* < 0.001; *M*_level 3_ = 887, *SD* =115, *t* (53) = −5.880, *p* < 0.001; *M*_level 4_ = 900, *SD =* 131, *t* (53) = −5.617, *p* < 0.001). However, there was no significant difference between the reaction times of level 2 and level 3 (*t* (53) = −2.363, *p* = 0.126), nor those of level 3 and level 4 (*t* (53) = −2.174, *p* = 0.213). Overall, as the difficulty levels moved higher, the reaction times increased.

### 3.2. Experiment 2 Results

#### 3.2.1. Learning Phase

Accuracy. Analysis showed that there was no significant group difference in the facial feedback conditions for accuracy, *F* (1, 49) = 0.382, *p =* 0.539, nor an interaction between facial feedback conditions and difficulty levels, *F* (3, 147) = 0.347, *p* = 0.792. However, there was a significant main effect of difficulty level, *F* (3, 147) = 20.041, *p* < 0.001, *η^2^_p_* = 0.290. Bonferroni post hoc comparisons indicated that the accuracy of level 1 (*M*_level 1_ = 0.841, *SD =* 0.125) was significantly higher than level 3 (*M*_level 3_ = 0.697, *SD =* 0.128, *t* (50) = 5.640, *p* < 0.001) and level 4 (*M*_level 4_ = 0.699, *SD =* 0.138, *t* (50) = 5.111, *p* < 0.001). The accuracy of level 2 (*M*_level 2_ = 0.802, *SD =* 0.107) was also significantly higher than level 3 (*t* (50) = 4.866, *p* < 0.001) and level 4 (*t* (50) = 4.461, *p* < 0.001). Nevertheless, there was no significant difference in accuracy between level 1 and level 2 (*t* (50) = 1.948, *p* = 0.361), nor between level 3 and level 4 (*t* (50) = −0.112, *p* = 1.000).

Reaction times. The reaction times of the in-group facial feedback conditions did not differ from those of the out-group facial feedback conditions, *F* (1, 49) = 0.261, *p* = 0.612. There was also no significant interaction, *F* (3, 147) = 0.257, *p* = 0.856. However, there was a significant main effect of difficulty level, *F* (3, 147) = 4.189, *p* = 0.007, *η^2^_p_* = 0.079. Bonferroni post hoc comparisons showed that the reaction times of level 4 (*M*_level 4_ = 1549, *SD =* 580) were significantly slower than level 2 (*M*_level 2_ = 1331, *SD =* 475, *t* (50) = −3.609, *p* = 0.004). However, there was no significant difference among the reaction times of level 1(*M*_level 1_ = 1365, *SD =* 589) and level 2 (*t* (50) = 0.619, *p* = 1.000), level 2 and level 3 (*M*_level 3_ = 1428, *SD =* 472, *t* (50) = −1.847, *p* = 0.435), and level 3 and level 4 (*t* (50) = −2.473, *p* = 0.103).

#### 3.2.2. Test Phase

Accuracy. Although there was no significant main effect of facial feedback conditions for accuracy, *F* (1, 48) = 1.090, *p =* 0.302, there was a significant main effect of difficulty level, *F* (3, 144) = 12.274, *p* < 0.001, *η^2^_p_* = 0.204, and there was a significant interaction between the facial feedback conditions and the difficulty level, *F* (3, 144) = 3.842, *p* = 0.011, *η^2^_p_* = 0.074. Bonferroni post hoc comparisons for the main effect of difficulty level found that the accuracy of level 1 (*M*_level 1_ = 0.719, *SD* = 0.165) was significantly higher than that of level 2 (*M*_level 2_ = 0.671, *SD* = 0.105, *t* (49) = 3.799, *p* = 0.002), level 3 (*M*_level 3_ = 0.633, *SD* = 0.099, *t* (49) = 3.700, *p* = 0.002), and level 4 (*M*_level 4_ = 0.573, *SD* = 0.171, *t* (49) = 3.497, *p* = 0.003). The accuracy of level 2 was slightly higher than that of level 3 (*t* (49) = 2.577, *p* = 0.064), and the accuracy of level 3 was significantly higher than that of level 4 (*t* (49) = 2.844, *p* = 0.024). Overall, accuracy gradually decreased from level 1 to level 4. 

In addition, a simple effect analysis of interaction showed that the accuracy of the in-group facial feedback condition (*M*_in-group_ = 0.768, *SD* = 0.160) was significantly higher than that of the out-group facial feedback condition (*M*_out-group_ = 0.675, *SD* = 0.159) at level 1, *t* (48) = 2.057, *p =* 0.045. Beyond that, there was a trend towards a difference, with it not reaching statistical significance between the in-group (*M*_in-group_ = 0.701, *SD =* 0.110) and out-group (*M*_out-group_ = 0.644, *SD* = 0.094) facial feedback condition at level 2, *t* (48) = 1.998, *p* = 0.051 (see Figure 5). However, there was no significant difference between the in-group (*M*_in-group_ = 0.641, *SD* = 0.118) and out-group (*M*_out-group_ = 0.626, *SD* = 0.080) facial feedback conditions at level 3 (*t* (48) = 0.535, *p* = 0.595) and level 4 (*M*_in-group_ = 0.536, *SD* = 0.169; *M*_out-group_ = 0.607; *SD* = 0.094; *t* (48) = −1.497, *p* = 0.141) (see Figure 5). Correspondingly, there were significant differences in the accuracy of the four difficulty levels in the in-group facial feedback condition, but no significant difference in the out-group facial feedback condition. In the in-group condition, the accuracy of level 1 was significantly higher than that of level 2 (*t* (23) = 3.713, *p* = 0.004); the accuracy of level 2 was significantly higher than that of level 3 (*t* (23) = 2.798, *p* = 0.040); and the accuracy of level 3 was significantly higher than that of level 4 (*t* (23) = 3.887, *p* = 0.005).

Reaction times. Similar to the learning phase, the main effect of facial feedback conditions was not significant for reaction times in the test phase, *F* (1, 48) = 1.195, *p* = 0.280, and there was no interaction between the facial feedback conditions and difficulty levels, *F* (3, 144) = 0.272, *p* = 0.845. Nevertheless, there was a main effect of difficulty level, *F* (3, 144) = 30.848, *p* < 0.001, *η^2^_p_* = 0.391. Bonferroni post hoc comparisons for the main effect of difficulty level showed that the reaction times of level 1 (*M*_level 1_ = 840, *SD* = 115) were significantly faster than level 2 (*M*_level 2_ = 858, *SD* = 129, *t* (49) = −3.288, *p* = 0.010), level 3 (*M*_level 3_ = 878, *SD =* 127, *t* (49) = −5.331, *p* < 0.001), and level 4 (*M*_level 4_ = 902, *SD =* 140, *t* (49) =−6.228, *p* < 0.001). The reaction times of level 2 were significantly faster than level 3 (*t* (49) = −4.179, *p* = 0.001) and level 4 (*t* (49) = −6.540, *p* < 0.001). The reaction times of level 3 were significantly faster than those of level 4 (*t* (49) = −4.804, *p* < 0.001).

## 4. Discussion

This study used the implicit rule-learning task paradigm and conducted two new experiments to examine how in-group and out-group facial feedback affect the implicit rule-learning task performance of different difficulty levels. In Experiment 1, the implicit rule-learning task included two phases, the learning phase and the test phase. The participants were asked to indicate whether the combinations of three cards complied with an implicit rule, and there was facial feedback after their responses. The participants were randomly assigned to the in-group facial feedback group or out-group facial feedback group, with feedback from the East Asian faces and Western faces being received separately. Setting in/out-group facial feedback as a between-subject variable rather than a within-subject variable allowed us to distinguish the different contributions of the two types of facial feedback to implicit learning. The results from Experiment 1 indicated that although there were significant main effects of difficulty level in the learning and test phase, there was neither a significant main effect of social feedback group nor a significant interaction between the difficulty and the social feedback group. In Experiment 1, as the participants were presented with only one type of facial feedback (East Asian or Western faces), their perception of the difference between in-group and out-group facial feedback may not be highlighted. 

Therefore, we designed Experiment 2 to further examine this. In Experiment 2, the implicit rule-learning task was the same as in Experiment 1, but before the group assignment, the participants were shown all the feedback from the in-group faces and the out-group faces. Then, they were randomly assigned to one of the groups and informed that they would receive only one type of feedback throughout the experiment. The results of the test phase in Experiment 2 showed that at difficulty level 1, the accuracy of the in-group facial feedback group was significantly higher than that of the out-group facial feedback group. The results suggested that out-group facial feedback had a negative impact on implicit rule learning at the lowest difficulty level. In addition, there was a significant difference in accuracy for the four difficulty levels in the in-group facial feedback group, while there was no difference in accuracy in the out-group facial feedback group.

### 4.1. Effect of Social Feedback on Implicit Rule Learning

The results indicated that in-group and out-group social feedback played a role in the implicit rule-learning task performance at different difficulty levels. The in-group facial feedback group showed a better learning effect when the tasks were easy. However, when the tasks became difficult, the out-group facial feedback group maintained the learning effect, which was embodied in the fact that the accuracy of the out-group facial feedback group did not significantly decrease as the task difficulty increased. According to previous studies, the implicit rule-learning task was cognitively demanding, and social feedback increased cognitive and attentional load during learning [10,20]. In the in-group facial feedback condition, since in-group faces were perceived as more familiar than out-group faces [21], individuals could pay less attention to in-group faces and invest more cognitive resources in the task, resulting in higher sensitivity to task difficulty. Therefore, the accuracy rate of the task changed with the difficulty, showing gradually decreasing performance as the difficulty increased. In the out-group facial feedback condition, since the out-group faces had more racial features than the in-group faces [28] and the out-group members were perceived as more threatening than the in-group ones [29], individuals paid more attention to the out-group faces and were then not sensitive to the task difficulty. Therefore, the accuracy rate did not change with the difficulty, with it showing stable performance. This possible mechanism could explain the higher accuracy of the in-group facial feedback group than that of the out-group facial feedback group at the lowest difficulty level. 

### 4.2. Comparison of the Two Experiments

Whether the participants were shown both the in-group and out-group facial images before the experiment was the main difference between Experiment 1 and Experiment 2. The results indicated that the influence of in-group and out-group facial feedback on implicit rule learning at different difficulty levels was only shown in the situation where the participants saw both the in-group and out-group faces before the experiment. One probable reason was that people were likely to pay more attention to the information of the facial expression than the characteristics of the group identity if they did not realize the social categorizations, but when they saw both the in-group and out-group faces, the cognition of group identity may have been highlighted, which may have changed the allocation of cognitive resources and then have had an influence on the implicit rule learning. The results were similar to the findings in terms of racial stereotypic beliefs [25], proving that the activation of cognition of in-group and out-group members was not an automatic consequence but a cognitive process affected by the existing goals or acquired information.

### 4.3. Limitations and Future Directions

Several limitations of the present study need to be noted. Regarding the selection of participants, most of the participants were female college students. Therefore, it was unknown whether the unbalanced gender ratio affected the task performance of implicit learning based on social feedback. Considering the large number of female teachers in primary and secondary schools [27] and the objective limitations of the image database (male East Asian faces were not abundant), the division of in-group and out-group facial feedback was only based on female racial faces. Future studies can further divide more specific in-group and out-group facial feedback according to the characteristics of participants, such as dividing by age or gender, so that such studies will have higher ecological validity. In addition, although facial expressions are common form of feedback, voice and body posture have a greater advantage in conveying information to large numbers of people over longer distances [30], such as in a situation where a teacher needs to provide voice feedback to all of the students in a large classroom. Thus, future studies can also compare the effect of in-group and out-group feedback on implicit rule learning through different nonverbal social feedback channels. For the measurement of implicit rule learning, only behavioral experiments were used in this study to explore the mechanism of implicit rule learning, which may lead to a lack of precision. EEG technology, such as event-related potentials (ERPs), have been proven to be effective in obtaining unbiased evidence of implicit rule learning [10]. In the future, EEG technology can be used to investigate the influence of different types of social feedback on implicit rule learning. 

Moreover, the present study was conducted on Chinese participants but not on other ethnic groups. Moreover, only the participants, but not the researchers, were blinded to the aim of the current study. In addition, the current study employed just one type of negative facial feedback, i.e., sad expression. Other types of negative feedback (e.g., frowning or angry expressions) may lead to different results. Notably, the current study was an initial study of implicit learning with social feedback, and we did not investigate the relationship between psychological and personality tests and implicit learning with social feedback.

In sum, future research can balance the gender ratio, expand the age range of participants, refine the division of in-group and out-group feedback, select more different feedback materials, employ a double-blind experimental design, use advanced technology such as EEG, and add psychological and personality tests and more types of emotional feedback to explore the influence of more types of social feedback on implicit learning and the corresponding neural mechanism in more ethnicities. 

### 4.4. Practical Implications

This study is the first to investigate the role of in-group and out-group social feedback in implicit learning and has made some practical contributions. First, because implicit learning is possible in an educational setting [31] and is preferable for language skills acquisition [32], improving motor actions [33,34,35], learning music [36], and completing social cognitive tasks [37], the effect of in-group and out-group facial feedback on implicit learning should be considered in educational practice, especially in the fields of international educational services. International schools and training institutions could assign in-group and out-group teachers according to the difficulty levels of learning. For example, it is better to assign in-group teachers instead of out-group teachers to provide feedback when students are learning skills or knowledge with simple rules or structures. Second, in an international educational environment, the student’s cognition of group identity and the feeling of being threatened by out-group teachers are likely to be activated, which may lead to a higher cognitive load during learning and testing. An effective way to prevent this negative influence is not to emphasize the social group identities of foreign teachers but to highlight their teaching roles.

## 5. Conclusions

This study aimed to compare the effects of in-group and out-group facial feedback on implicit rule learning using a new implicit–intentional learning task in two experiments. We found that when the contrast of group identities was highlighted, out-group feedback reduced the accuracy of the least difficult task; on the contrary, when the social group identities were not emphasized, there was no accuracy difference between the out-group and in-group feedback conditions.

## Figures and Tables

**Figure 1 behavsci-13-00963-f001:**
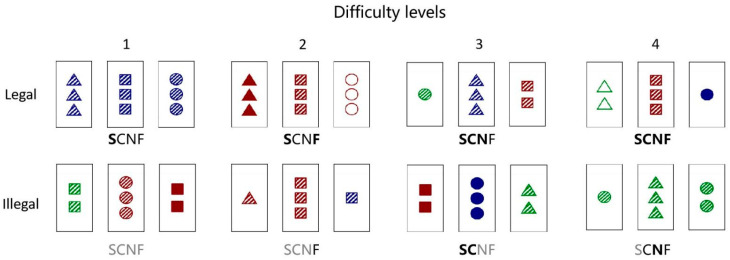
Example card triad stimuli for the implicit–intentional learning task. Black slim letters indicate that the elements of the dimension in three cards were the same, black bold letters were coded for completely different elements of the dimension in three cards, and gray letters showed that the elements of the dimension in the card triad were partially different.

**Figure 2 behavsci-13-00963-f002:**
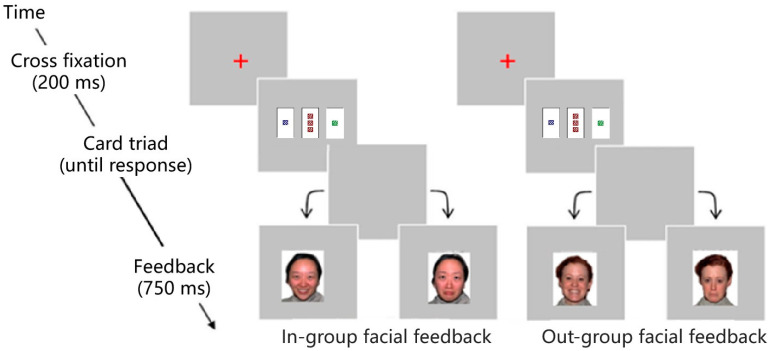
The conditions of in-group (left) and out-group (right) facial feedback.

**Figure 3 behavsci-13-00963-f003:**
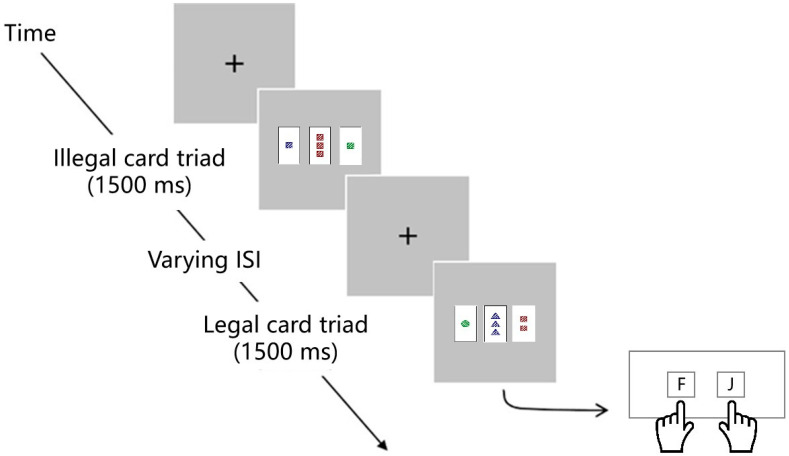
Experimental procedure in the test phase.

**Figure 4 behavsci-13-00963-f004:**
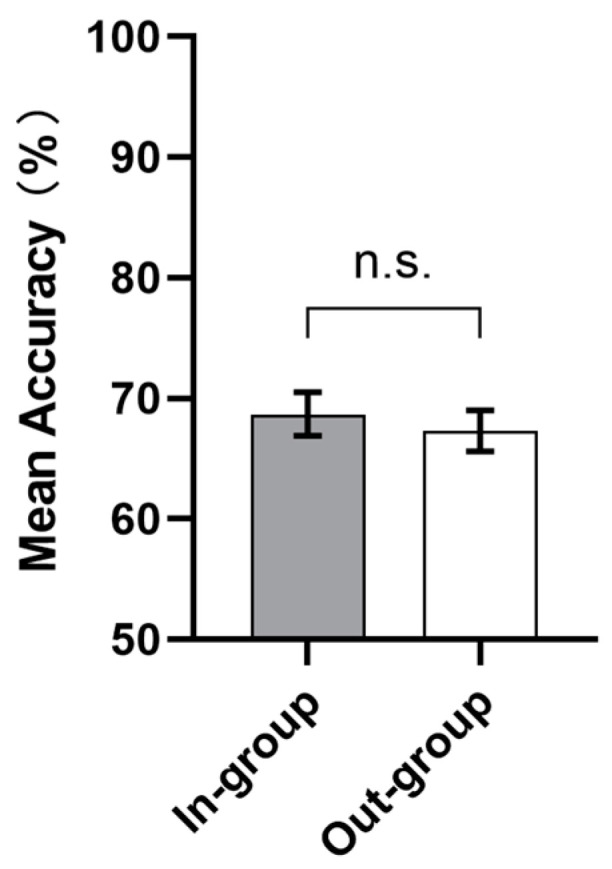
The results from the test phase of Experiment 1 showed that there was no significant difference between the in-group and out-group facial feedback conditions in terms of accuracy. Error bars represent the standard error of the mean accuracy. n.s. *p* ≥ 0.1.

**Figure 5 behavsci-13-00963-f005:**
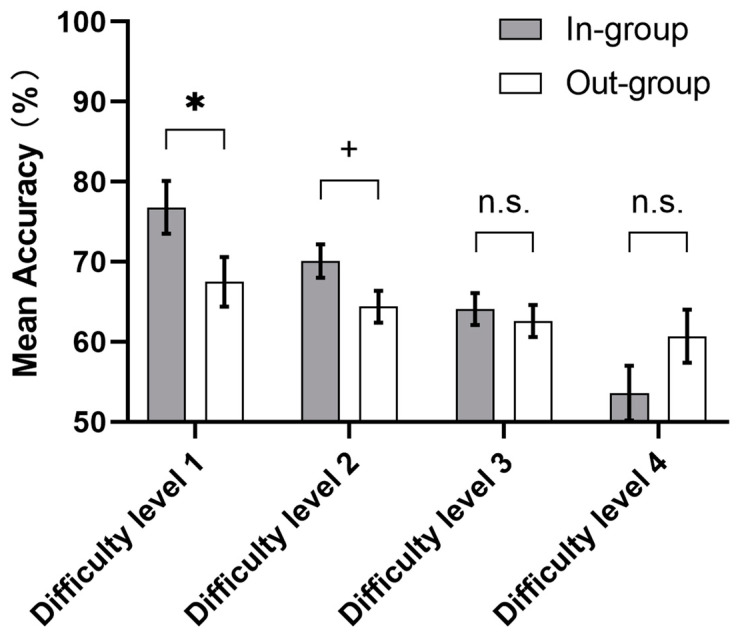
The results from the test phase of Experiment 2 showed that there was a significant interaction between the facial feedback conditions and difficulty levels in terms of accuracy. Error bars represent the standard error of the mean accuracy. + *p* < 0.1, * *p* < 0.05, n.s. *p* ≥ 0.1.

## Data Availability

The data presented in this study are available on request from the corresponding author.

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
