# Peer review of "The Role of In-Group and Out-Group Facial Feedback in Implicit Rule Learning"

_behavsci, 2023, doi:10.3390/bs13120963_

Round 1
Reviewer 1 Report
Comments and Suggestions for Authors
INTRODUCTION
The introduction to the manuscript is too long. Most of this needs to be moved to the discussion section. Your discussion section is in fact shorter then the intro. Please only include a primer and background on the topic. And then go straight into the objective, hypothesis, and aims of the present study.
Consider organizing the information in a logical sequence. Start with a general overview of implicit learning and its relevance, narrow down to the specifics of social feedback and its potential impacts, and then introduce the particular contexts and variables of your study.
While the context related to China is helpful, consider whether it can be more briefly and effectively integrated into the narrative, maintaining a global perspective.
Ensure that all terms are defined concisely and effectively. For example, clarify terms such as “in-group and out-group facial feedback” at the first mention.
Emphasize the potential implications of the findings in a more global context, focusing on the advancements in learning processes and educational methodologies, rather than focusing primarily on the Chinese context.
METHODS:
What was the recruitment methods for the participants in the study. How were individuals randomized, and were the researchers blinded? In the cohort in experiment 2 by what means was the survey distributed to the students.
Explain why specific stimuli were chosen, clarifying the relevance and appropriateness of the chosen stimuli for the study.
Provide more detail on the creation and selection process of card triads and facial expressions.
Specify whether the stimuli used in the practice phase are significantly different from those in the experimental phase and if they had any effect on the results.
Clarify the instructions given to participants for each task and how these were communicated.
Offer more detail on any post-hoc analyses conducted, including the rationale and any adjustments made for multiple comparisons.
Comments on the Quality of English Language
Only minor editing needed.
Reviewer 2 Report
Comments and Suggestions for Authors
The paper investigates how in-group and out-group facial feedback impact different difficulty levels of implicit rule learning. Specifically, by comparing the implicit learning through one type of facial feedback vs learning through both in-group and out-group facial feedback authors report that accuracy of the in-group facial feedback group was significantly higher than that of the out-group facial feedback group. Additionally, the authors report that when the contrast of group identities was highlighted, out-group feedback reduces accuracy of the least difficult task; on the contrary, there is no accuracy difference between out-group and in-group feedback conditions. Overall, it is a well-designed study with interesting findings. However, I have some major concerns and suggestions that I would like the authors to consider/address to improve the quality of the manuscript.
· The introduction is well-written and I don’t have any major concerns/suggestions.
· In Experiment 1 results, the results show that there is no significant impact of facial feedback (in-group vs out-group). The only significant findings observed in the learning phase and test phase if logical (not novel) where difficult tasks were completed with low accuracy and required higher reaction times.
· In Experiment 2, the learning phase results as the same as the learning phase result in Experiment 1. However, in the test phase, authors report a significant interaction effect between facial feedback and difficulty levels. Did the authors perform same interaction effect in prior cases (Experiment 1 and Experiment 2 Learning Phase?). I ask because nothing is mentioned regarding the same (significant vs non-significant interaction effects for Experiment 1 and Experiment 2 Learning Phase. Also, can the authors update current Figure 4 and present it same as how Figure 5 is presented.
· Figure 5, I am a bit surprised that the there is no-significant difference for Difficulty 4 between In vs Out group.
· Major Concern – The primary/major findings of the study “The results showed that only in Experiment 2 but not Experiment 1, there was a significant interaction effect in the accuracy of tasks between difficulty level and group. At the least difficulty level, the learning accuracy of the in-group facial feedback group was significantly higher than that of the out-group facial feedback group, whereas it did not happen at two highest levels of difficulty. In conclusion, when the contrast of group identities was highlighted, out-group feedback reduces accuracy of the least difficult task; on the contrary, there is no accuracy difference between out-group and in-group feedback conditions.” are supported by marginal significant differences between groups (in-group vs out-group) during difficulty 1 and 2 where observed p-value = 0.045 and p-value = 0.051. Moreover, the findings are inverted during Difficulty although no p-values are reported. I don’t feel comfortable with the main findings drawn/reported from the study as both values are marginal. Hence I would suggest to modify the results and tone down the language regarding the claim as these could be potentially confounded by other variables.
Comments on the Quality of English Language
No major concerns
Reviewer 3 Report
Comments and Suggestions for Authors
In this study the Authors investigate the influence of social feedback on implicit learning, specifically the role of in-group vs. out-group facial expressions, concluding that accuracy is higher for in-group feedback with low difficulty tasks
The subject is worth exploring, especially in consideration of the increasing internationalization of education, both in China and worldwide. However, several major issues need to be considered:
- one main question is: why did the Authors choose to compare in- and out-group faces in different subjects and not directly in the same subjects? Applying both conditions in the same subjects could rule out several possible confounding factors (see below: different educational level, different familiarity with out-group individuals). The Authors should explain the reason of their choice, and if not adequately justified, they should mention this as a limitation
Section 1.3:
- the sentence “which is similar to symbols” is not justified by adequate references, moreover, if true, it would contradict previous studies, which differentiate facial expressions from symbols: the Authors should either delete it, or explain it exhaustively.
- the sentence “a typical example…”, defining in-group interactions as “typical”, appears to contradict the aim of the study, denying the occurrence of out-group interaction in education: either delete it, or change it so that the possibility of both in- and out-group interaction is suggested.
- the frequent references to what happens in China as apparently exclusive of Chinese culture should be generalized, as a mix of different ethnicities is very common (and getting even more so) in several instances worldwide.
- the sentence “and played an important role in the performance of implicit rule learning [7].” is ambiguous in that it suggests a relevant role in learning performance, which is not justified by the findings in ref 7; the Authors should either delete it, or rephrase and explain it.
- the last sentence of section 1.3 (“Thus, we designed two sections…”) deals with the present study and seems to conclude the introduction; the Authors should either delete it or modify it to integrate it in section 1.4.
Section 2.1:
- whereas it is understandable that there are sometimes less subjects in the test phase (drop-outs), how is it possible that in Experiment 1 for the out-groups in the test phases there is one female subject more than in the learning phase? What did the Authors test in this subject, if there has been no learning phase?
- the enrollment is described differently for the two experiments: is it a real difference, or just a different description? For Exp 2 were the subjects also students? The Authors should describe the enrollment more accurately, and specify possible differences between the two experiments and their reasons, for instance, had the subjects the same level of familiarity with out-group individuals? Also, please note that educational level and social background could potentially interfere with the conditions of the experiments, subjects with higher levels of education, or living in large cities, could have had more opportunities to be acquainted with international counterparts (teachers, instructors). I suggest that the Authors recall their subjects and present them with appropriate questionnaires to assess these aspect; then they can use the data to either rule out these confounding effects (if no significant differences are present in these respects among the groups), or take them into consideration in the analysis (using the obtained scores as regression parameters).
- did the Authors propose questionnaires, such as personality/psychological tests? It would be interesting to correlate the results, for instance, with some traits such as empathy (IRI), or approach-avoidance trait (BIS-BAS). While the Authors recall the subjects for the previous point, they could present them with these (and other?) questionnaires as well.
Section 2.2:
- Feedback sad expression, not frowning: is there a special reason for this choice? Are there elements to hypothesize that sad and frowning faces could lead to different results?
Section 2.3.2
- How were the images presented to the subjects, with which explanation/instruction? Were they made aware of the aim of the study, or lead to believe that they could see either (both?) kinds of faces? This is very briefly dealt with in Discussion (section 4), but the appropriate place is here, in Methods, and it should be explained in greater detail
Section 3.2.2
- what do the Authors mean by “significant marginal difference”? It appears that these results could be better described as a “trend towards a difference, not reaching statistical significance”: is this what they meant? The Authors should either change or clarify.
Section 4.3
- the possibility to perform analogous studies in other ethnicities as respondents should be considered as a limitation of the present study and/or a possible future development
Comments on the Quality of English Language
Language should be checked throughout the manuscript for several mistakes and stylistic faults, such as (just a few examples here!):
“refers to that people acquire”: “refers to the fact that people acquire”
“feedback, i.e., in-group (East Asian)”: “feedback, i.e., either in-group (East Asian)”
“facial feedback become more and more common”: became? has become? ...?
Round 2
Reviewer 2 Report
Comments and Suggestions for Authors
Thanks for addressing all my comments.
Comments on the Quality of English LanguageNo major issues. Minor grammatical changes can be made.
Reviewer 3 Report
Comments and Suggestions for Authors
Please notice that lines mentioned in the following comments refer to the track-changes version of the manuscript
- Response 2: The Authors should give the same explanations they provide here about the experimental protocol also in the manuscript, although in more concise form (e.g., 3-4 lines); I suggest in Discussion, after line 402, after “Western faces separately” and before “Results from Experiment 1…”
- Response 7: Beyond the fact that the paragraph “in addition, … implicit learning” is repeated twice (see also comment below), the new sentence is extremely obscure, namely, what do the Authors mean by “reminding affecting”? what do they mean by “farthing learned”? In the present form the whole sentence is basically incomprehensible
- Response 10: The Authors should briefly mention the issue of psychological and personality tests in section 4.3, Limitation and future directions
- Response 11: The Authors’ explanations about using sad faces instead of frowning faces may be reasonable, however, they are the ones who suggest a frown for negative feedback (Section 1.3, Line 78: “… with a smile or frown [10,17-20]”). Also, generally speaking, frowning appears as a much more common negative reaction to wrong answers. I am not asking the Authors to re-do the whole study, but they should at least acknowledge this issue and the possibility that different negative feedback may give different results (or maybe not? Are there elements in this sense in the literature?)
The manuscript should be thoroughly and carefully checked, as several mistakes and inaccuracies are present; they give an overall impression of sloppiness which hinders the correct appreciation of the manuscript:
- Lines 61-64 are the same line, repeated several times
- The two paragraphs at lines 96-107 and 108-119 are in reality the same paragraph, though with different changes marked
- Lines 209 and 210 are the same line, repeated
Comments on the Quality of English LanguageSeveral language faults are still present; here just a few examples (not an exhaustive list, please check the manuscript thoroughly):
- Line 71: “It found that” should be changed to “These Authors found that” or similar
- Line 80: “In-group facial feedback referred to the facial expression from individuals within their groups.” Should be “In-group facial feedback refers to the facial expression of individuals within the same ethnic group as the observer”, and correspondingly for the following sentence “out-group…".
- Line 88: the sentence “educational resources on the world” is obscure
- Line 123: “affecting” probably means “affects”? otherwise the sentence is incomprehensible
- Line 124: “was” should be “is”
- Line 145: “the social media platform” seems to indicate only one platform, therefore it should not be followed by “e.g.”. Otherwise, it should be “on social media platforms (e.g., WeChat group)”
- Line 159: it should read “as in Experiment 1”
- Line 163: “both two phases” should be “both phases”
- Line 232: “… complied with a rule, i.e., whether they were either legal…”
Round 3
Reviewer 3 Report
Comments and Suggestions for Authors
The paper has been considerably improved
Please note a typo, line 45 "that strongly influences..." should be "... that strongly influence..."